# The Enrichment of miRNA-Targeted mRNAs in Translationally Less Active over More Active Polysomes

**DOI:** 10.3390/biology12121536

**Published:** 2023-12-18

**Authors:** Tingzeng Wang, Shuangmei Tian, Elena B. Tikhonova, Andrey L. Karamyshev, Jing J. Wang, Fangyuan Zhang, Degeng Wang

**Affiliations:** 1Department of Environmental Toxicology, and The Institute of Environmental and Human Health (TIEHH), Texas Tech University, Lubbock, TX 79416, USA; tingzeng.wang@gmail.com (T.W.); shuangmei.tian@ttu.edu (S.T.); 2Department of Cell Biology and Biochemistry, Texas Tech University Health Sciences Center, Lubbock, TX 79430, USA; elena.tikhonova@ttuhsc.edu (E.B.T.); andrey.karamyshev@ttuhsc.edu (A.L.K.); 3Department of Cancer Biology and Genetics, James Comprehensive Cancer Center, Wexner Medical Center, The Ohio State University, Columbus, OH 43210, USA; wang.12230@osu.edu; 4Department of Mathematics and Statistics, Texas Tech University, Lubbock, TX 79416, USA; fangyuan.zhang@ttu.edu

**Keywords:** microRNA (miRNA), polysome, translationally less-active light polysome, more-active heavy polysome, polysome profiling, DICER1, 3′-untranslated regions (UTR)

## Abstract

**Simple Summary:**

MicroRNAs (miRNA) inhibit the translation and enhance the degradation of their target mRNAs, but only moderately. Paradoxically, miRNAs and their target mRNAs are polysome-associated, which should protect mRNAs from degradation. To mechanistically solve the paradox, we performed comparative translationally less-active light and more-active heavy polysome profiling of human cells. Isogenic mutant cells incapable of mature miRNA production due to a disrupted DICER1 gene were used as the background model. The enrichment of miRNA-targeted mRNAs in light- over heavy-polysome was observed. That is, though polysome-associated, miRNA-targeted mRNAs are enriched in the translationally less-active polysome complexes. This enrichment reconciles the seemingly contradictive miRNA regulatory activities, their moderateness and the polysome association.

**Abstract:**

miRNAs moderately inhibit the translation and enhance the degradation of their target mRNAs via cognate binding sites located predominantly in the 3′-untranslated regions (UTR). Paradoxically, miRNA targets are also polysome-associated. We studied the polysome association by the comparative translationally less-active light- and more-active heavy-polysome profiling of a wild type (WT) human cell line and its isogenic mutant (MT) with a disrupted DICER1 gene and, thus, mature miRNA production. As expected, the open reading frame (ORF) length is a major determinant of light- to heavy-polysome mRNA abundance ratios, but is rendered less powerful in WT than in MT cells by miRNA-regulatory activities. We also observed that miRNAs tend to target mRNAs with longer ORFs, and that adjusting the mRNA abundance ratio with the ORF length improves its correlation with the 3′-UTR miRNA-binding-site count. In WT cells, miRNA-targeted mRNAs exhibit higher abundance in light relative to heavy polysomes, i.e., light-polysome enrichment. In MT cells, the DICER1 disruption not only significantly abrogated the light-polysome enrichment, but also narrowed the mRNA abundance ratio value range. Additionally, the abrogation of the enrichment due to the DICER1 gene disruption, i.e., the decreases of the ORF-length-adjusted mRNA abundance ratio from WT to MT cells, exhibits a nearly perfect linear correlation with the 3′-UTR binding-site count. Transcription factors and protein kinases are the top two most enriched mRNA groups. Taken together, the results provide evidence for the light-polysome enrichment of miRNA-targeted mRNAs to reconcile polysome association and moderate translation inhibition, and that ORF length is an important, though currently under-appreciated, transcriptome regulation parameter.

## 1. Introduction

MicroRNAs (miRNAs), initially discovered in 1993 in *Caenorhabditis elegans* [1] and later found evolutionarily conserved in metazoan species, are vital regulators of mRNA translation and degradation. Major components of the three segments of the miRNA pathway—biogenesis, targeting and regulatory actions—are generally known [2]. Defects in the pathway are frequent etiological factors for human diseases, such as cancer [2,3,4,5,6,7,8,9].

Canonical miRNA biogenesis starts from pri-miRNA transcripts’ synthesis viaRNA polymerase II or III. The Drosha RNase III enzyme, in partnership with the RNA-binding protein DGCR8, digests the pri-miRNA into pre-miRNAs, with one pri-miRNA often producing multiple (up to six) pre-miRNAs. The pre-miRNAs travel, via the nucleocytoplasmic shuttle Exportin-5 (XPO5), from the nucleus into the cytoplasm. The DICER1 RNase III enzyme, in partnership with the TARBP2 RNA-binding protein, processes the pre-miRNA further into a mature 22-nucleotide long miRNA duplex [10,11,12]. The duplex is then loaded onto the Argonaute (AGO) proteins, and the passenger strand is removed.

There are non-canonical biogenesis pathways as well [13,14]. Mirtrons and 7-methylguanine-capped (m7G)-pre-miRNAs both circumvent the pri-miRNA synthesis and the Drosha processing steps; mirtrons are produced directly from mRNA introns [13,15]; (m7G)-pre-miRNAs are directly exported to the cytoplasm through Exportin-1 (XPO1) [16]. Additionally, one evolutionally conserved miRNA, mir-451, is DICER-independent [17]. It is processed via Drosha from endogenous short hairpin RNA (shRNA) transcripts. The pre-miRNA is not sufficiently long to be a DICER substrate. Instead, it binds directly to AGO proteins to complete its maturation within the cytoplasm. In DICER1 knockout cells, canonical miRNAs are still detected, but at markedly reduced levels (median 0.058% of wild type cell levels); pre-miRNAs are loaded directly onto AGO and trimmed at the 3′ end, yielding miRNAs from the 5′ strand (5p miRNAs) [18]. Nevertheless, this knockout experiment further confirmed the importance of Dicer1 in mature miRNA production.

The miRNA–AGO complex then exerts regulatory actions onto target mRNAs via base-pairing between miRNA seed sequences, which in human is only six–eight nucleotides long, and cognate binding sites; other binding efficacy factors are also involved [19]. With the PIWI domain binding to the 5′-end of loaded miRNA, and the PAZ domain to the 3′-end, AGO orients the miRNA for base pairing with target mRNAs. At the same time, the AGO protein disassociates from TARBP2 and DICER1 [12] and recruits the p-body (processing body) scaffold protein TNRC6A/B/C (Trinucleotide Repeat Containing 6), forming the core of the miRNA-targeting machinery and bridging upstream miRNA biogenesis to downstream regulatory effectors. The TNRC6s, in turn, recruit downstream effectors—general translation inhibition and/or mRNA destabilization machinery such as the CCR4–NOT and PAN2–PAN3 complexes.

There remain gaps in our understanding of miRNA-regulatory actions. One paradox is the miRNA–AGO association with the active poly-ribosome complex, commonly termed polysome, despite their mRNA translation inhibition and degradation activities. The polysome association was observed in the very early miRNA studies. Subsequently, numerous studies reproduced this observation, though the extent of polysome association relative to association with mono-ribosome and ribosome-free cytoplasmic fractions varies from study to study and from miRNA to miRNA [20,21,22,23,24,25]. Additionally, miRNA-mediated translation inhibition and mRNA degradation are very moderate. We do not have a mechanistic understanding of the moderateness and the polysome association.

This study examines, to the best of our knowledge, for the first time, the polysome association by the comparative polysome profiling analysis of a human cell line and its isogenic DICER1 knockout mutant. We simultaneously analyzed translationally less-active light and more-active heavy polysomes. Our results provide evidence for the enrichment of miRNA-targeted mRNAs in the light polysomes, reconciling polysome-association, miRNA suppressive regulatory actions and their moderateness. We also showed that transcription factors and protein kinases are the top two groups of most enriched mRNAs.

## 2. Materials and Methods

### 2.1. Cell Culture

HCT116 is a widely used human colon cancer cell line [26,27]. The wild type HCT116 and DICER1 knockout HCT116 cells were generous gifts from Dr. Bert Vogelstein of Johns Hopkins University that created the isogenic mutant cells and showed mature miRNA production deficiency in them [26,28]. They were grown in McCoy’s 5A medium supplemented with 10% Fetal Bovine Serum (FBS) and 1% Penicillin-Streptomycin at 5% CO_2_, 37 °C.

### 2.2. Evolutionarily Conserved miRNA Binding Site Count

As previously reported, to alleviate the high noise issue in miRNA binding site prediction due to the short site length, we used evolutionarily conserved human miRNA binding sites in this analysis. The set of conserved sites from the TargetScan database 8.0 was downloaded in March 2023 [29,30]. For each mRNA, the number of unique miRNAs that have conserved 3′-UTR sites was computed as the miRNA binding site count.

### 2.3. Polysome Profiling

Polysome profiling was performed as previously described [31,32,33]. Briefly, we treated the cells with 100 μg/mL cycloheximide for 15 min at 37 °C and 5% CO_2_, followed by two washes with cold DPBS. We lysed the cells with lysis buffer (10 mM Tris pH 7.5, 100 mM KCl, 5 mM MgCl_2_, 1 mM DTT, 0.5% Triton X-100, 1x protease inhibitor cocktail (EDTA-free), 200 units/mL of RNase inhibitor), followed by centrifugation to pellets and removed the nucleus. Collected cytoplasmic lysates were loaded on top of a 10 to 60 percent sucrose gradient, followed by centrifugation in a Beckman SW41 rotor at 390,000× *g* at 4 °C for two hours. The gradient was fractioned into 25 fractions. As previously reported [33], we assigned 10-mer or more, i.e., with 10 or more ribosomes on the mRNA, as heavy polysomes, and 2-to-9-mer as light polysomes. The heavy and light polysome fractions, identified based on the OD_260_ profile of the fractionation process, were collected for total RNA isolation. Mutant polysome profile was similar to wild type profile until 9-mer, but differed from it afterward.

The total RNA samples were processed for NGS sequencing analysis by BGI America. The samples were first treated with DNase I to degrade any possible DNA contamination. Then, the oligo(dT) magnetic beads were used to enrich the mRNAs. The mRNA was fragmented into short fragments (~200 bp). Then, the first cDNA strand was synthesized by using random hexamer-primer. Buffer, dNTPs, RNase H and DNA polymerase I were then added to synthesize the second strand. The double-stranded cDNA was purified with magnetic beads, followed by end reparation and 3’-end single nucleotide A (adenine) addition. Finally, sequencing adaptors were ligated to the fragments, and the fragments were enriched by PCR amplification. Following quality control and quantification, the libraries were analyzed on a BGI America DNBseq sequencer. The BGI America sequencing team pre-processed the raw sequencing reads to filter out low quality reads and remove the multiplexing barcode sequences, and then provided clean reads in FASTQ format. As previously discussed [33], the NGS dataset is publicly available at the NCBI GEO database (accession number GSE134818).

The sequencing reads in the comparative polysome RNA-seq dataset were aligned to the human reference genome (hg38) with the STAR alignment software (version 2.4.1d), followed by gene expression level calculation with the HTSeq-count software (version 2.0.3) [34].

### 2.4. The Kyoto Encyclopedia of Genes and Genomes (KEGG) and Gene Ontology (GO) Gene Sets

We downloaded the KEGG and GO functional gene sets from the Molecular Signatures Database (MSigDB) version 2023.1 at the Gene Set Enrichment Analysis (GSEA) website (https://www.gsea-msigdb.org/gsea/index.jsp, accessed on 30 July 2023) [35,36,37]. The KEGG gene set contained 186 gene sets. The GO molecular function (MF) set contained 1772 sets.

### 2.5. Statistical Analysis

We used the R open source statistical software (version 4.2.2) installed on a Mac Pro desktop computer for statistical analysis. Outlier identification, Student *t*-test, F-test, descriptive statistical parameter calculation, correlation coefficient calculation, linear regression, LOESS regression, random sampling and other statistical procedures were all performed with this R software.

### 2.6. LOESS Regression and the Adjustment of the mRNA Abundance Log-Ratio with Open Reading Frame (ORF) Length

The adjustment is made with the loess function of the R open source statistical software (version 4.2.2) with default settings, i.e., span equal to 0.75. The light to heavy polysome mRNA abundance log-ratio was the response variable, and the ORF length in log-scale was the predictive variable. The residuals of the resulting regression model, i.e., the difference from the predicted values, was used as the adjusted log-ratio. The R-squared (R^2^) was calculated to quantify the portion of variance explained by the LOESS regression, though it was not as frequently used as in other parametric regression analyses.

## 3. Results

### 3.1. The Enrichment of MiRNA-Targeted mRNAs in Translationally Less-Active Light Polysomes Suggested by the Comparative WT–Mutant Cell Analysis

MiRNAs are well documented to inhibit the translation activity of their target mRNAs. We asked whether this inhibition is reflected in the target-mRNA distribution among the polysomes by using our light and heavy polysome profiling datasets for both WT and isogenic mutant (DICER1 knock-out) (MT) HCT116 cells. We also used, as in our previous works, the evolutionarily conserved 3′-UTR miRNA binding site count as the degree to which a mRNA is regulated by miRNAs [32,33].

A shift in miRNA-targeted mRNAs from light to heavy polysomes in mutant cells was observed (Figure 1A). The MT-to-WT mRNA abundance log-ratio in heavy polysomes (log_2_(MT_H/WT_H) steadily increased as the miRNA binding site count increased (Figure 1A, black datapoints). For light polysomes (log_2_(MT_L/WT_L), the trend was much weaker (Figure 1A, red datapoints). That is, the comparative analysis of WT and MT cells suggested that miRNA-targeted mRNAs are enriched in light polysomes in WT cells. The DICER1 knockout disrupts the enrichment, releasing the mRNAs into heavy polysomes, in MT cells.

### 3.2. The Inconsistency with Raw Light to Heavy Polysome mRNA Abundance Log-Ratios of WT Cells

Encouraged by the results, we directly examined light to heavy polysome mRNA abundance log-ratios (log_2_(WT_L/WT_H)) in WT cells. However, we obtained seemingly inconsistent results. This log-ratio (Figure 1A, blue datapoints) was not much miRNA-binding-site-count-dependent; unlike the MT-to-WT heavy polysome abundance ratio (log_2_(MT_H/WT_H), it exhibited at most a dubious/marginal correlation with the binding site count. The analysis is also shown as comparative boxplots of this log-ratio of miRNA-targeted mRNAs (>50 3′-UTR miRNA binding sites) and untargeted mRNAs (no binding site) (Figure 1B). As previously observed, miRNA-targeted mRNAs seem to be under tighter regulation, i.e., the log-ratio exhibiting a much lower level of dispersion (WT-cell F-test *p*-value: 0.019). MiRNA-targeted mRNAs seem to have higher log-ratios than untargeted mRNAs (Figure 1B, “WT > 50 sites” versus “WT No Site”). However, the difference is very small, though statistically significant (*t*-test *p*-value: 0.041) and absent in MT cells (Figure 1B, “MT > 50 sites” versus “MT No Site”).

### 3.3. ORF Length as a Determinant of Light-to-Heavy Polysome mRNA Abundance Log-Ratio and an Effect of DICER1 Disruption on the Log-Ratio

To address this discrepancy, we surveyed other factors that might affect the log_2_(WT_L/WT_H) log-ratio. And, mRNA ORF length came to our attention; with all else the same, the log-ratio should correlate well with mRNA ORF length, as a longer ORF can accommodate more translating ribosomes. Thus, we tested whether ORF length was the interfering factor responsible for the poor correlation between the log_2_(WT_L/WT_H) log-ratio and miRNA binding site count in WT cells.

The result is shown as scatter-plots (log-ratio versus ORF length) in Figure 2, confirming ORF length as a determinant of the log-ratio in both WT (Figure 2A) and MT cells (Figure 2B). As discussed in Materials and Methods, log-ratio versus ORF length LOESS regressions (the blue data points in Figure 2) were used to illustrate the trend of decreasing log-ratios as ORF length increases. The trend is already clear in WT cells (Figure 2A). It seems even stronger in MT cells (Figure 2B); the data-points seem to fit much more tightly into the trend, exhibiting lower levels of dispersion from the LOESS regression. The R^2^ value increased from 0.215 to 0.362. Additionally, the DICER1 disruption seemed to reduce the log-ratio value range. The MT-cell datapoints (Figure 2B) are concentrated in narrower ranges than that of the WT-cell datapoints (Figure 2A). This is confirmed by the comparative boxplots of WT and MT cell log-ratio values (F-test value: <2.2 × 10^−16^) (Figure 3A). Thus, the DICER1 disruption has a clear effect on the log-ratio.

### 3.4. The ORF-Length-Adjusted Light to Heavy Polysome mRNA Abundance Log-Ratio

The correlation with ORF length suggests that, in order to better reveal the miRNA-mediated regulatory activity on mRNA distribution in light versus heavy polysomes, there is a need to adjust the mRNA abundance log-ratio with ORF length. The need is illustrated by mRNAs with more than 50 miRNA binding sites, which are highlighted as red data points in Figure 2A. Collectively, these mRNAs exhibit high ORF-length-adjusted log-ratios, i.e., positive LOESS regression residuals (the vertical difference between each data point and predicted value) (Figure 2A). In other words, their raw log-ratios are higher than what would be expected by the LOESS regression.

Thus, we used the LOESS regression residuals as ORF-length-adjusted log-ratio (Figure 3). The adjusted log-ratio confirmed quantitatively the observation described above: that the data points fit more tightly into the LOESS regression in mutant cells (Figure 2B) than in WT cells (Figure 2A). As illustrated by a comparative boxplot of the adjusted log-ratios (Figure 3A), the WT cells exhibit higher levels of dispersion than mutant cells (F-test *p*-value: <2.2 × 10^−16^). Thus, miRNA regulation rendered the mRNA abundance log-ratio less dependent upon ORF length in WT cells, and its disruption strengthened the dependency in mutant cells.

Not surprisingly, comparative boxplots exhibit an obvious difference between mRNAs with >50 miRNA binding sites and mRNAs with no such sites (Figure 3B). The difference is much larger than that shown in Figure 1B; the statistical significance given by the *t*-tests improved dramatically, with the *p*-value decreasing from 0.041 to 1.8 × 10^−6^ (Figure 1B and Figure 3B). Thus, the ORF length adjustment led to an improved differentiation of the two groups of mRNAs. Once again, miRNA-targeted mRNAs exhibited a lower level of dispersion (F-test *p*-value: 0.019) (Figure 3B). To further confirm the statistical significance of this observation, we randomly picked 1000 samples of mRNAs from the transcriptome with the same sample size of 49 and calculated their mean adjusted log-ratios. None of the sample means (maximum 0.41) are even close to the mean (0.55) of targeted mRNAs (Figure 3B). Thus, the higher ORF-length-adjusted log-ratios of miRNA-targeted mRNAs are extremely unlikely to be random events.

The effect of ORF length is further illustrated in a genome-wide manner with another LOESS regression. This regression incorporated the miRNA binding site count as an additional predictive variable, i.e., log-ratio versus ORF length + log_2_(binding-site-count) (Figure 4). The predicted log-ratio increased as the binding site count increased, but the trend became much clearer only when viewed in the context of ORF length—the *x*-axis (Figure 4).

### 3.5. MiRNA-Targeted mRNAs Tend to Have Longer ORFs

Figure 4 also showed that mRNAs with high binding site counts (dark red data-points) tend to concentrate in long ORF length ranges of the plot, i.e., to the right of the vertical bar that denotes the mean ORF length. A systematic analysis confirmed this observation (Figure 5). As the binding site increases, the mean ORF length steadily increases, passing the genome-wide mean ORF length denoted by the horizontal line at the binding site count of 5 (Figure 5A). The data points fit almost perfectly into the trend until the binding site count reaches 20, but then exhibit higher levels of scatteredness, likely due to lower numbers of mRNAs with given binding site counts and thus reduced statistical power. Many parameters follow the so-called scale-free distribution [38,39,40,41,42,43,44,45,46]. The MiRNA binding site count is one such parameter [32,33,47]; as miRNA binding site count increases, the number of mRNAs with the count decreases exponentially. Additionally, a similar trend was also observed with the binding site count normalized by ORF length, i.e., the site count per KB of ORF length (Figure 5B).

### 3.6. Incorporating ORF Length into the Analysis

Given these results, we incorporated ORF length into our analysis, i.e., using ORF-length-adjusted light to heavy polysome mRNA abundance log-ratio (see Materials and Methods). This led to a dramatic enhancement of the dependency upon the 3′-UTR miRNA binding site count (Figure 6A). Unlike the raw log-ratio, the adjusted log-ratio steadily increases as the binding site count increases. The data points fitted almost perfectly into the trend at low binding site count ranges. They then exhibited progressively higher levels of scatteredness, likely due to the exponentially decreasing numbers of mRNAs with given site counts. Nevertheless, overall, the trend persists.

On the other hand, there are many non-miRNA translation regulation mechanisms. The ORF-length-adjusted mRNA abundance log-ratio of WT cells alone is not sufficient to exclude one or more of these mechanisms as the causes of the correlation with binding site count—the so-called confounding effect in statistical analysis. To address this issue, we subtracted from it the ORF-length-adjusted log-ratio of the isogenic mutant cells. That is, we calculated the changes of the adjusted log-ratio caused by the DICER1 disruption, thus filtering out the potential effects of other regulatory factors. As shown in Figure 6B, after this second adjustment, the log-ratio still correlates well with miRNA binding site count; a nearly perfect linear correlation was observed.

### 3.7. The Enrichment of Protein Kinase and Transcription Factor in Light-Polysome-Enriched mRNAs

To identify the cellular functions that are most regulated by mRNA light-polysome-enrichment, we performed a gene-set enrichment analysis of the double-adjusted mRNA abundance log-ratio with the KEGG functional gene sets and the Gene Ontology (GO) molecular function (MF) gene sets [48,49,50]. The top 20 most enriched gene sets are listed in Table 1. For the KEGG sets, 8 of the top 20 terms are cellular signaling pathways, with the MAP Kinase signaling pathway ranked at number one (Table 1, red text). Not surprisingly, 16 of the top 20 GO MF terms are related to the components of cellular signaling: protein kinase, transcription factor (TF) and other signaling molecules (Table 1, purple, green and blue text, respectively). The six TF-related GO MF terms are ranked at number 1, 2, 3, 6, 13 and 20 (Table 1, green text). The five protein kinase-related terms are ranked at number 4, 5, 7, 12 and 15 (Table 1, purple text).

## 4. Discussion

This study used comparative polysome profiling analyses of the WT HCT116 cell and its isogenic DICER1 knockout mutant to shed light onto miRNA-mediated translation inhibition activity. We separated the polysomes into translationally less active, i.e., with fewer ribosomes on the mRNA, light fractions and more active fractions. By analyzing the light and the heavy polysomes separately, we observed that miRNAs retain their target mRNAs in the light polysomes, thus inhibiting, though not completely shutting off, their translation. To the best of our knowledge, this is the first report of this phenomenon based on a genome-wide study.

Notably, to clearly reveal the light-polysome-enrichment of target mRNAs, mRNA ORF length needs to be incorporated into our analysis. Firstly, ORF length is a determinant of the light to heavy polysome mRNA abundance log-ratio, though it is rendered less powerful by miRNA-mediated regulatory activities in WT cells than in the mutant cells. The log-ratio needs to be viewed in this context to fully differentiate target and non-target mRNAs. Secondly, target mRNAs are not randomly distributed in the transcriptome in terms of their ORF lengths. Instead, they tend to have longer ORFs, i.e., coding for longer proteins, than non-target mRNAs.

The mechanistic and functional underpinning of ORF length as an important transcriptome regulatory parameter is not yet completely understood. However, it is conceivable to argue that the metabolic and energetic overhead cost of translation might be an important factor [51,52]. Synthesizing the amino acids levies the metabolic overhead. During elongation, adding each amino acid to the nascent polypeptide chain consumes one ATP and two GTP molecules. The ATP is used to conjugate the amino acid to the tRNA, i.e., to charge the tRNA; one GTP is used for binding the charged tRNA to a ribosome A-site, and the other for subsequent ribosome translocation towards the next codon. In a word, translation is an expensive process; the longer the ORF, the more expensive it becomes. Consequently, it is not surprising to observe mRNAs with longer ORFs under tighter miRNA regulations. Additionally, there is a myriad of other mRNA regulatory mechanisms, such as the large number of RNA-binding proteins (RBP) [53,54,55,56]. It will be interesting to examine whether some of them tend to target mRNAs with long ORFs as well.

Perhaps not coincidentally, proteins encoded in the human genome are longer (median 431 amino acids) than those in the budding yeast genome (median 358 amino acid), which are in turn longer than those in *E. coli* genome (median 277 amino acid) (http://book.bionumbers.org/how-big-is-the-average-protein/, accessed on 30 July 2023). It becomes interesting to investigate whether longer protein length is generally applicable to the metazoan genomes. It is possible that the miRNA regulatory system and its mode of mRNA regulation emerged to meet the needs of a tighter regulation of mRNAs for longer proteins in metazoan organisms during biological evolution.

The light-polysome-enrichment of miRNA-targeted mRNAs explains key seemingly contradictive observations of miRNA regulatory actions. It reconciles the paradox of the polysome association and translation inhibition. It also helps to explain the modesty of miRNA-mediated regulatory activities, that is, target mRNA degradation and translation inhibition. The enrichment is equivalent to low, but not completely shut-off, translation activities. On the other hand, the association with light polysomes protects these target mRNAs from the fate of the quick degradation of ribosome-free mRNAs. Consistently, the co-translational degradation of the target mRNAs has been observed, reconciling their polysome-association and enhanced degradation. It has also been shown that miRNA-targeted mRNAs are degraded through a different pathway [57].

However, the functional advantages this regulatory system confers to the cell remain to be fully illuminated; it seems wasteful that the cells expend critical metabolic and energetic resources to produce these mRNAs, but then render them translationally inhibited under enhanced degradation pressure [58,59,60]. Perhaps, we can borrow insights from computer design in operational latency mitigation. Computers and the cells are frequently analogized to each other [61,62,63]. The computer information retrieval from the hard drive to memory, and then to CPU caches, is strikingly similar to the gene expression process [62,64,65,66]. The retrieval process is much slower than CPU cycle execution; the CPU might have to stay idle for extended periods of time while waiting for information for the next cycle. To minimize this latency, various principles are implemented for the speculative retrieval of information prior to the CPU request [64,65,66]. We are investigating the applicability of these principles in miRNA-mediated regulatory actions, as gene expression in human cells is a slow and latency-causing process.

Our results also point to future studies. This study compared steady-state mRNA polysome distributions between WT and isogenic DICER1 knockout mutant cells. Though the difference correlated well with miRNA binding site count, it is possible that some of the observations are secondary effects of the DICER1 knockout [67,68,69]. To fully distinguish direct and secondary DICER1 knockout effects, the DICER1 gene would need to be introduced back into the mutant cells with an inducible expression vector. Comparative polysome profiling studies can then be performed at multiple time points following the activation of Dicer1 expression. The restoration of the light polysome retention of miRNA-targeted mRNAs should occur prior to other secondary effects. Additionally, we are incorporating other parameters, such as the miRNA affinity/binding-energy to the targets, into the analysis.

## 5. Conclusions

In summary, we present evidence for enrichment of miRNA-targeted mRNAs in the translationally less-active light polysomes, advancing our understanding of miRNA-mediated translation inhibition. ORF length is an essential parameter in such analysis of miRNA regulatory activities, in that it is needed to fully reveal the light polysome enrichment and that miRNA-targeted mRNAs tend to have longer ORFs. That is, ORF/protein length is an important, but currently under-appreciated, factor in transcriptome regulation.

## Figures and Tables

**Figure 1 biology-12-01536-f001:**
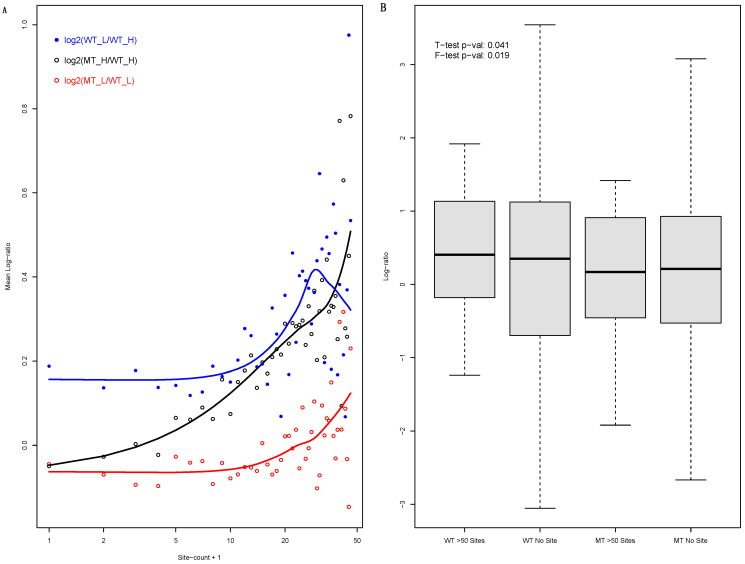
MiRNA-targeted mRNAs’ shift toward translationally more-active heavy polysomes in isogenic DICER1 mutant cells, i.e., suggesting enrichment in less-active light polysomes in WT cells, and its inconsistence with the raw light to heavy polysome mRNA abundance log-ratio in WT cells. (**A**)**:** Scatter plots of mRNA abundance log-ratios versus mRNA 3′-UTR miRNA binding site counts (WT: wildtype cells; MT: mutant cells; H: heavy polysome; L: light polysome). Mean log-ratio of genes with corresponding binding site counts were plotted. Binding site count is in log scale, with one added to each count to avoid the log_2_(0) error. LOESS regression trend curves are also shown. (**B**)**:** Comparative boxplots of light to heavy polysome abundance log-ratios of mRNAs with more than 50 binding sites and those with no identified site in WT cells (“WT > 50 Sites” and “WT No Site”) as well as MT cells (“MT > 50 Sites” and “MT No Site”). The *t*-test and F-test *p*-values for WT cells are shown.

**Figure 2 biology-12-01536-f002:**
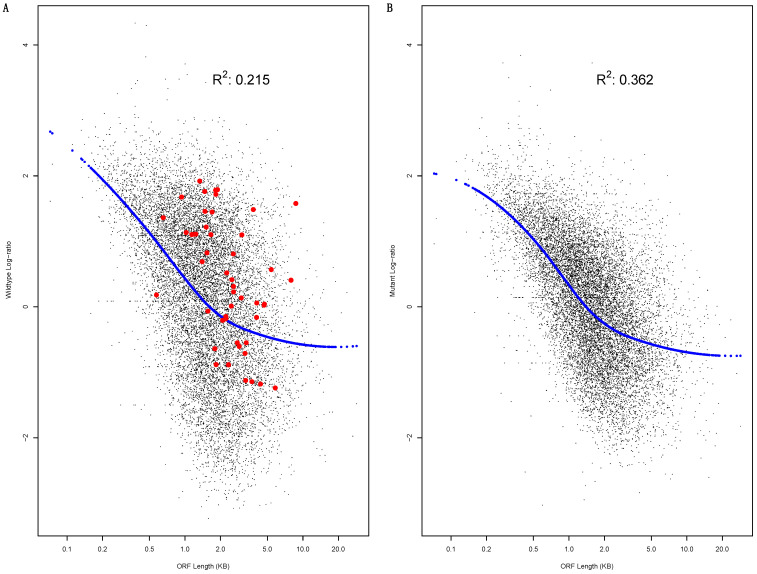
Open reading frame (ORF) length as a determinant of light to heavy polysome mRNA abundance log-ratio. A scatter plot of the log-ratio versus ORF length is shown for WT (**A**) and mutant cells (**B**). LOESS regressions (log-ratio versus ORF length) are used to illustrate the relationship, with predicted values shown as blue datapoints. The R^2^ value of the regression is also shown. In (**A**), the genes with more than 50 miRNA binding sites in their mRNA 3′-UTR are highlighted as red data points.

**Figure 3 biology-12-01536-f003:**
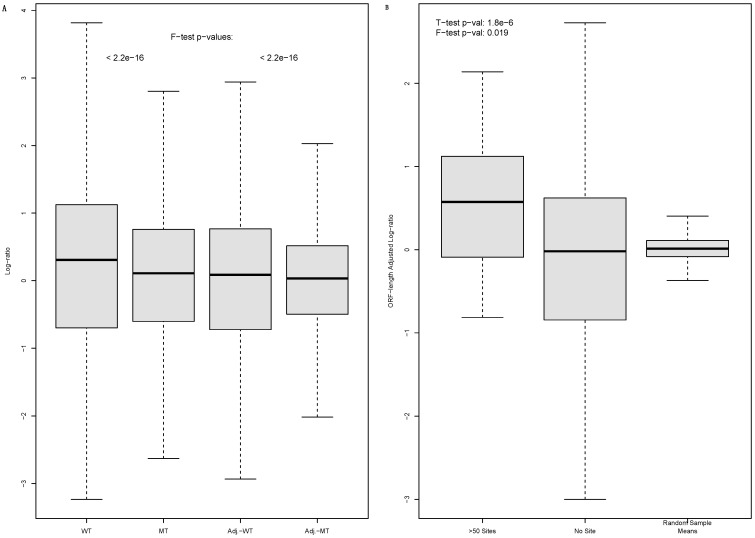
Comparative boxplots of light to heavy polysome mRNA abundance log-ratio. (**A**): A boxplot to compare raw log-ratio (WT and MT) and ORF-length-adjusted log-ratio (Adj.-WT and Adj.-MT), i.e., the residuals of LOESS regressions in Figure 2, both illustrating reduced dispersion in mutant cells. That is, DICER1 disruption reduces raw log-ratio’s value range and makes ORF length a more powerful determinator of the log-ratio in mutant cells. (**B**): Boxplots of the adjusted log-ratio to compare mRNAs with more than 50 3′-UTR miRNA binding sites, mRNAs with no sites and the means of random mRNA samples in WT cells. A more significant difference between miRNA-targeted and untargeted mRNAs is observed than in Figure 1B; the t- and F-test *p*-values are shown.

**Figure 4 biology-12-01536-f004:**
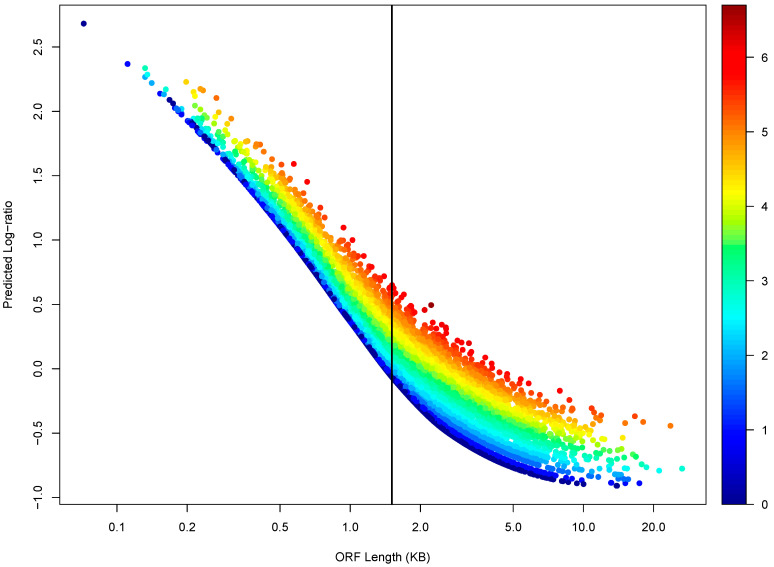
Relationship between light to heavy polysome mRNA abundance log-ratio and miRNA binding site count viewed in the context ORF length. A LOESS regression (log-ratio versus ORF length plus log_2_(binding site count)) was performed for WT cells. A scatter plot of the predicted log-ratio versus ORF length is shown, with the data points color-coded by the log_2_(binding site count) values. The vertical bar denotes mean ORF length.

**Figure 5 biology-12-01536-f005:**
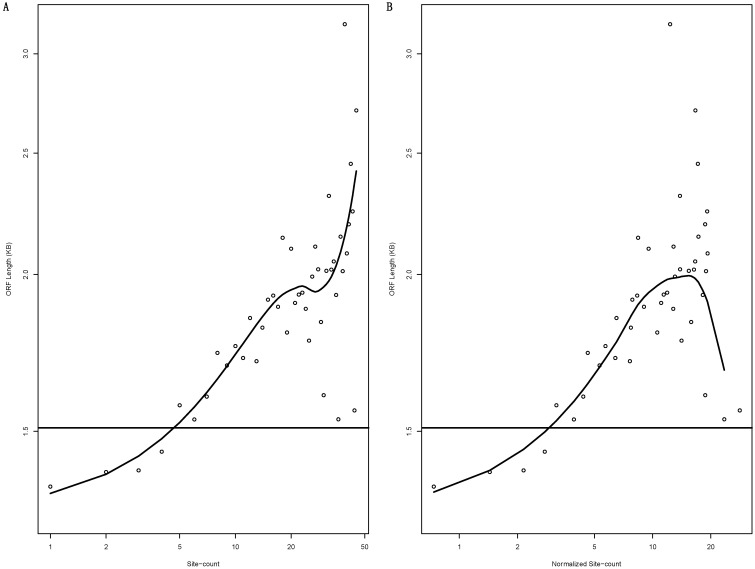
MiRNA-targeted mRNAs tend to have longer ORFs. The scatter plots display mean ORF length versus corresponding 3′-UTR miRNA binding site counts (**A**) or the site counts normalized by ORF length in KB (**B**), with both axes in log scale. The smooth curve denotes the trend computed by a LOESS regression. The horizontal bar denotes the mean log_2_(ORF length).

**Figure 6 biology-12-01536-f006:**
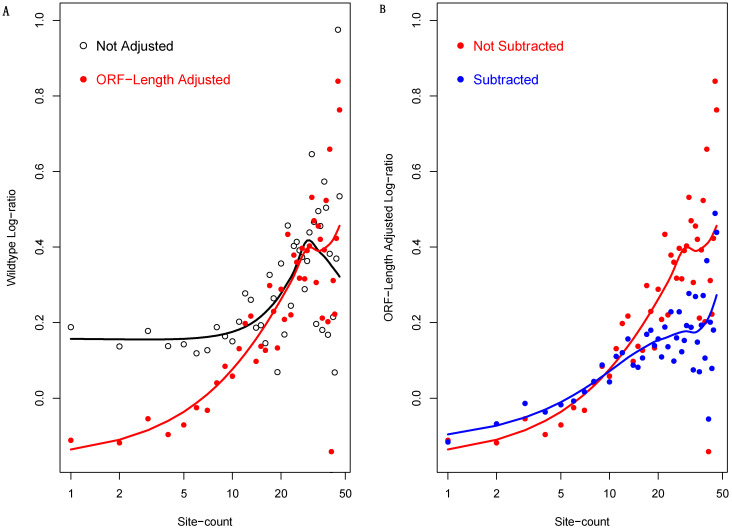
ORF length adjustment improves correlation between light to heavy polysome mRNA abundance log-ratio and miRNA binding site counts. For *x*-axis, all site counts were increased by one to avoid the log_2_(0) error. Panel (**A**) compares raw log-ratio and ORF-length-adjusted log-ratio in WT cells. Panel (**B**) compares the ORF-length-adjusted log-ratio and the adjusted log-ratio subtracted by the adjusted log-ratio of mutant cells. Upon the subtraction, a nearly perfect linear correlation is observed.

**Table 1 biology-12-01536-t001:** Top 20 most light-polysome-enriched KEGG and GO molecular function (MF) terms. KEGG terms for cellular signaling pathways are in red text. GO MF terms related to transcription factors are in green text, terms for protein kinase activities in purple text and terms for other signaling molecules in blue text.

Rank	KEGG	GO MF
1	MAPK_signaling_pathway	DNA_binding_transcription_factor_activity
2	chronic_myeloid_leukemia	cis_regulatory_region_sequence_specific_DNA_binding
3	neurotrophin_signaling_pathway	sequence_specific_DNA_binding
4	renal_cell_carcinoma	protein_kinase_activity
5	ERBb_signaling_pathway	protein_serine_threonine_kinase_activity
6	glioma	transcription_regulator_activity
7	pancreatic_cancer	protein_serine_kinase_activity
8	phosphatidylinositol_signaling_system	G-protein_coupled_receptor_activity
9	glycosaminoglycan_biosynthesis_keratan_sulfate	molecular_transducer_activity
10	melanogenesis	voltage_gated_potassium_channel_activity
11	chemokine_signaling_pathway	nucleoside_triphosphatase_regulator_activity
12	non_small_cell_lung_cancer	kinase_activity
13	melanoma	DNA_binding_transcription_activator_activity
14	neuroactive_ligand_receptor_interaction	delayed_rectifier_potassium_channel_activity
15	hedgehog_signaling_pathway	receptor_tyrosine_kinase_binding
16	acute_myeloid_leukemia	UDP_glycosyltransferase_activity
17	GnRH_signaling_pathway	GTPase_activator_activity
18	Fc_epsilon_RI_signaling_pathway	phosphatidylinositol_3_kinase_regulator_activity
19	Fc_gamma_R_mediated_phagocytosis	phosphatase_regulator_activity
20	colorectal_cancer	DNA_binding_transcription_repressor_activity

## Data Availability

The NGS dataset, as previously described [33], is publicly available in the NCBI GEO database (accession GSE134818).

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
