# Peer review of "The Enrichment of miRNA-Targeted mRNAs in Translationally Less Active over More Active Polysomes"

_biology, 2023, doi:10.3390/biology12121536_

Round 1

Reviewer 1 Report

Comments and Suggestions for Authors

I thank the authors for submitting the article titled "Enrichment of miRNA-Targeted mRNAs in Translationally Less Active over More Active Polysomes" to Biology. The article is generally well-written; scientifically sound however, I do have some recommendations to make prior to publication.

Comments:

1.     In the present study, wild-type HCT116 and DICER1 knockout HCT116 cells were used for polysome profiling. Previous studies have reported that the preference for low-high polysome occupancy of miRNAs depends on the affinity of the microRNAs for their mRNA targets (Molotski, N. and Soen, Y., 2012). Does the affinity of microRNAs play any role in light-heavy polysome enrichment? Please clarify.

Author Response

Thank you for your encouragement. 

In this work, we do not take specific properties of individual mRNA, such as the affinity/binding-energy to target sites used by Molotski and Soen in the 2012 study, into consideration. However, we added a statement in the discussion that such properties will be considered in our future larger-scaled studies.

Reviewer 2 Report

Comments and Suggestions for Authors

Unfortunately, I consider an inappropriate self-citation by the last author, that should be solved before publication.

Author Response

Thank you for your time and efforts. Our responses to your comment are listed below:

1: We appreciate that you pointed out that one previous publication (Molecular Toxicology Protocols, 2000) seemed cited twice. This is due to spelling errors in one author's name and affiliation in the original publication. We are not sure that the mistakes have been corrected in all published versions. So it becomes necessary to cite both the article and its notice of correction. The two are essentially one publication. If this is taken into consideration, our submission is following the journal self-citation policy. This re-submission follows the policy regardless.

2: The rest of our self-citations discuss our contribution to the occurrence of the scale-free distribution in biology and its relevance to miRNA biology, feedback control of the AGOs and miRNA targeting activity, as well as our works in crafting a functional context to interpret this important but seemingly wasteful regulatory mechanism. 

Reviewer 3 Report

Comments and Suggestions for Authors RNA interference, a mechanism of transcription regulation discovered several decades ago, continues to be an intriguing field of study, with some aspects of transcription regulation by miRNAs still warranting further elucidation. In their comprehensive research, the authors have undertaken a commendable effort to bridge this knowledge gap. They employed a dual approach, conducting both light- and heavy-polysome profiling on human cells to assess the abundance of microRNAs (miRNAs), which is an original and highly pertinent topic within the realm of molecular biology.  

The methodology employed in this study stands out for its elegance and rigor. The authors have diligently utilized appropriate methods to gather their data, and their experimental design is robust. Notably, they observed a significant enrichment of miRNA-targeted mRNAs in the light-polysome fraction, shedding light on an aspect of translational control. This finding hints at a crucial revelation: miRNA-targeted mRNAs tend to be associated with translationally less active polysome complexes. This, in turn, paves the way for a deeper understanding of the intricate regulation of gene expression by miRNAs.

One particularly noteworthy aspect of this study is the emphasis placed on the length of the open reading frame (ORF) as a critical parameter in transcriptome regulation. This finding not only enhances our comprehension of miRNA-mediated regulation but also adds a new layer to our understanding of how cells finely tune gene expression.

In terms of scientific rigor, the results presented in the manuscript firmly support the working hypothesis. The conclusions drawn by the authors align seamlessly with the evidence they have meticulously presented throughout the paper. Furthermore, the citation of references is judiciously done, lending credibility to their work.

In summary, this manuscript represents a significant contribution to our knowledge of the regulatory role of miRNAs in translation. It is well-crafted, scientifically sound, and highly relevant to the field. I wholeheartedly recommend its acceptance in its current form.

Author Response

We greatly appreciate your time and efforts in reviewing our manuscript. Your encouragement will motivate us to further contribute to this important research area.

Reviewer 4 Report

Comments and Suggestions for Authors

In this manuscript, the authors performed polysome profiling of WT and DICER1-KO in HCT116 cell lines. By integrative analysis, the authors identified enrichment of miRNA-target mRNAs in the light polysomes. Meanwhile, there are several concerns that might weaken the manuscript:

1. The authors claimed that the results provide direct evidence for enrichment of miRNA-target mRNAs in the light polysomes. This is an over-statement based on current results. The miRNA-targeted mRNAs were inferred from the evolutionarily conserved 3’-UTR miRNA binding site count. The authors didn't perform experiments to identify the miRNA-mRNA interaction in the control and treatment groups, the authors didn't perform experiments to identify the miRNA abundance in the control and treatment groups either. As such, the results couldn't be regarded as direct evidence for enrichment of miRNA-target mRNAs in the light polysomes.

2. The data analysis procedures should be detailed in the methods section, especially the log-ratio and the correction with ORF length, and isogenic mutant.

3. In figure 5, it shows mRNAs with longer ORF length tend to have more miRNA binding site count. Did the author consider to use site count per KB to normalize the ORF length among all mRNAs?

4. Line 254 “miRNA-target mRNAs tend to have longer ORFs”. It seems a little bit contradictory: on one hand, miRNA-target mRNAs tend to be enriched in the light polysomes. On the other hand, mRNAs with longer ORFs tend to be deposited in the heavy polysomes. How to reconcile this statement?

Comments on the Quality of English Language

The quality of English language is OK.

Author Response

Thank you for your constructive suggestions, which guided following item-by-item improvement:

1. The word "direct" has been deleted, though we would like to point out that we acquired the WT and Dicer mutant HCT116 cells from Dr. Bert Vogelstein. In a comparative study of the same cells, they showed that mature miRNA production is compromised in the mutant cells. Wa now state this in the Methods section.

2. We have modified the methods section accordingly.

3. Figure 5 now has panel A and B. Panel A is figure 5 in our original submission. Panel B uses miRNA binding site counts normalised by ORF length as you suggested.

4. We regret that our writing of the first part of the Results section is a bit confusing. This part now states more clearly the shift of miRNA-targeted mRNAs toward heavy polysomes in DICER1 mutant cells and its seemingly inconsistent with the row light- to heavy-polysome mRNA abundance log-ratio in WT cells. LOESS regression versus the ORF length corrected the effects of ORF length and the bias of miRNA-target mRNAs (longer ORF length), thus solving the inconsistency.

Reviewer 5 Report

Comments and Suggestions for Authors

In this study, the authors used polysome profiling to reveal that miRNA-targeted mRNAs, despite being associated with polysomes, are enriched in translationally less active polysome complexes. However, further analyses are needed before considering publication.

1. The authors compared the count of miRNA binding sites from light- to heavy-polysome mRNAs in WT and MT libraries and found that miRNA-targeted mRNAs are enriched in light polysomes in WT cells and shifted to heavy polysomes in MT cells. The reviewer has two main concerns:

Firstly, Figure 1A is somewhat complex and does not directly illustrate the pattern. Therefore, the reviewer suggests using a boxplot with points to visualize the site counts in light- to heavy-polysome mRNAs in WT and MT libraries and calculating the statistical significance of the differences.

Secondly, the reviewer noticed that the authors have used their previous generated only one library without replicates. It is crucial to clarify whether these results are derived from replicated samples and whether the observed pattern is consistent among replicate samples. Therefore, it is recommended to generate replicate libraries to ensure the robustness and reliability of the findings.

2. The authors compared the abundance of light- to heavy-polysome-related mRNAs in WT cells and found that miRNA-targeted mRNAs with over 50 binding sites had slightly higher abundance compared to mRNAs with no miRNA binding sites in the 3'-UTR. The reviewer is curious about whether there are statistically significant changes in MT cells. It is suggested to integrate this result into Figure 1B.

3. The authors observed that the trend of decreasing log-ratios as ORF length increases is evident in WT cells, and it appears stronger in MT cells, as shown in Figure 2A-B. The distributions in the two cases look quite similar, the reviewer suggests that it might be beneficial to compare and emphasize the pattern in WT and MT cells using an alternative method, such as a density line plot.

4. The reviewer noticed that the method description for the ORF-length adjusted log-ratio is missing. It is recommended to include this description in the methods section for clarity and completeness.

5. The reviewer is concerned that the results presented by the authors do not sufficiently exclude the possibility of random conditions. It is important to determine whether all the comparisons made in this paper show significant differences when compared to random conditions. To address this, it is strongly recommended to generate a random gene set and conduct comparisons with miRNA-targeted mRNAs to assess the statistical significance of the observed differences.

6. The reviewer is seeking a clearer understanding of the biological significance of the authors' study and its contributions to the research field. It is suggested that the authors emphasize how their results support and advance the existing research, providing a more explicit connection between their findings and their broader scientific implications. This will help clarify the importance of the study.

7. Other minor comments:

Introduction section: The authors should consider providing a more comprehensive introduction to the research progress in this field on the relationships between miRNA, miRNA targets, and polysomes to help readers understand the significance of the current study within the broader scientific landscape.

Lines 117-119: The authors have classified polyribosomes as light or heavy based on the criteria of assigning 10-mer or more ribosomes to mRNA as heavy polysomes and 2-to-9-mer as light polysomes. It is essential to provide a clear rationale for this classification method. Additionally, the authors should consider citing their previous work published in Nucleic Acids Research (NAR) in 2020 to support this classification approach.

Lines133-136: The description of data analysis process in the paper is relatively concise, and there is a need for more comprehensive explanations for each step. This should include specific details such as the software used for filtering out low-quality reads, the exact versions of the software, and the parameters employed in the process.

Lines146-150: The statistical methods in the paper should be described in more detail. It is important to include an explanation of the specific analyses that each statistical method is used for, the R packages employed, the analysis functions applied, and any parameters used.

Author Response

We greatly appreciate your efforts in reviewing our manuscript and raising these constructive suggestions for improving it. Our item-by-item responses  to your comments are as follows.

1 and 2: We regret that our writing of the first part of the Results section caused confusion. This part has been heavily modified. It now states more clearly the shift of miRNA-targeted mRNAs toward heavy polysomes in DICER1 mutant cells and its seemingly inconsistent with the row light- to heavy-polysome mRNA abundance log-ratio in WT cells shown in figure 1B. We pointed out that the difference shown in figure 1B, though significant based on a t-test, is too small. As suggested, the box plots for the MT cells is added to figure 1B.

        We appreciated that you pointed out that our NGS analysis did not use replicate. This is why our analysis compare large miRNA or mRNA groups, instead of individual miRNA or mRNA, between WT and MT cells. The replicate is much less of a concern, as bias/errors like cancel each other with big groups. We are working on NGS with replicates, and would update in future publications.

3: We added R-squared values, i.e., the percentage of variance explained by the LOESS regression, to the figure. WT cells show lower R-squared than MT cells. We are aware that R-squared is not frequently used in assessing LOESS/LOWESS regressions than parametric regressions, and state that it is used as just the percentage of variance explained by the LOESS regression.

4: The LOESS adjustment is added to the Methods section.

5: Where applicable, comparison with randomly selected/sampled sets is added into the Results section.

6: The Introduction and the Discussion sections have been updated.

7: We expand the discussion about the relationships among miRNA, target mRNA and polysome. We cited our NAR publication in the Methods section about light and heavy polysome. Additional details are added to the Methods section. 

Round 2

Reviewer 4 Report

Comments and Suggestions for Authors

The revised version of the manuscript has improved significantly. I recommend the publication of this manuscript.

Comments on the Quality of English Language

The quality of English language looks good to me.

Author Response

We greatly appreciate your help in improving our manuscript.

Reviewer 5 Report

Comments and Suggestions for Authors

The authors have implemented some modifications, but the review still finds the current paper lacking improvement compared to the previous version, so there is a need for further enhancement in the quality of this paper.

Lines 136-140: The description of the data analysis process is relatively concise. The reviewer suggests the inclusion of more comprehensive explanations for each step. Specific details, such as the software used for filtering out low-quality reads and removing multiplexing barcode sequences during the pre-processing stage of raw sequencing data, should be provided. Additionally, specifying the exact versions of the software and the parameters employed in the process would contribute to a clearer understanding.

Regarding the random iterations conducted by the author to calculate mean adjusted log-ratios, it is noted that none of the sample mean ratios closely align with the mean ratio of the targeted mRNAs. In light of this observation, the reviewer recommends not only expressing this finding in the written language but also integrating the randomized results into the relevant figure. It will contribute to a more comprehensive representation of the research outcomes.

Author Response

Thank you very much for your time and efforts. Our responses are below:

  1. The description of raw NGS-read pre-processing is moved to line 134, clarifying that the pre-processing was performed by the NGS company, BGI America. As this is a well-established NGS company, their approaches should be standard and reliable. We directly received clean high-quality reads in FASTQ format. The change is high-lighted as green text.
  2. A box plot of the mean log-ratios of the random samples is added to figure 3B. The addition is high-lighted as green text in the legend.